# Reduced affinity of calcium sensing-receptor heterodimers and reduced mutant homodimer trafficking combine to impair function in a model of familial hypocalciuric hypercalcemia type 1

Xiaohua Wang[1¤], James Lundblad[2,3], Stephen M. Smith[1,4] *

1 Division of Pulmonary and Critical Care Medicine, Oregon Health and Science University, Portland, Oregon, United States of America, 2 Division of Endocrinology and Diabetes, Oregon Health and Science University, Portland, Oregon, United States of America, 3 Section of Endocrinology and Diabetes, VA Portland Health Care System, Portland, Oregon, United States of America, 4 Sections of Pulmonary and Critical Care Medicine and Research & Development, VA Portland Health Care System, Portland, Oregon, United States of America

¤ Current address: Department of Pathology, Orlando Regional Medical Center, Orlando, Florida, United States of America
* smisteph@ohsu.edu

**Data Availability Statement:** All relevant data are within the paper and its Supporting information files.

## Abstract

Heterozygous loss-of-function mutation of the calcium sensing-receptor (CaSR), causes familial hypocalciuric hypercalcemia type 1 (FHH1), a typically benign condition characterized by mild hypercalcemia. In contrast, homozygous mutation of this dimer-forming G-protein coupled receptor manifests as the lethal neonatal severe hyperparathyroidism (NSHPT). To investigate the mechanisms by which CaSR mutations lead to these distinct disease states, we engineered wild-type (WT) and an exon 5-deficient disease-causing mutation, and transfected expression constructs into human embryonic kidney (HEK) cells. WT protein was mainly membrane-expressed whereas the mutant CaSR protein (mCaSR) was confined to the cytoplasm. Co-expression of WT CaSR directed mCaSR to the cell membrane. In assays of CaSR function, increases in extracellular $[Ca^{2+}]$ ($[Ca^{2+}]_o$) increased intracellular $[Ca^{2+}]$ ($[Ca^{2+}]_i$) in cells expressing WT CaSR while the response was reduced in cells co-expressing mutant and WT receptor. Untransfected cells or those expressing mCaSR alone, showed minimal, equivalent responses to increased $[Ca^{2+}]_o$. Immunoprecipitation experiments confirmed an association between mutant and wild-type CaSR. The affinity of the WT CaSR for calcium was three times greater than that of the heterodimer. The maximal functional response to $[Ca]_o$ was dependent on localization of CaSR to the membrane level and independent of homo- or heterodimerizations. In summary, these results suggest that heterodimerization of WT and mCaSR receptors, rescues the trafficking defect of the mutant receptors and also reduces the affinity of the WT-mutant heterodimer for $[Ca]_o$. In contrast, the homozygous mutants do not produce functional receptors on cell membrane. These data indicate how substantial differences between signaling of hetero-

**Funding:** Research reported in this publication was supported by the National Institute of General Medical Sciences of the National Institutes of Health (R01GM134110) and by U.S. Department of Veterans Affairs (BX002547) to SMS. The content is solely the responsibility of the authors and does not necessarily represent the official views of the National Institutes of Health or the U.S. Department of Veterans Affairs. The funders had no role in study design, data collection and analysis, decision to publish, or preparation of the manuscript.

**Competing interests:** The authors have declared that no competing interests exist.

and homodimeric mutants may lead to profound differences in the severity of disease in heterozygous and homozygous carriers of these mutations.

## Introduction

Familial hypocalciuric hypercalcemia Type 1 (FHH1) is an autosomal dominant disorder characterized by hypercalcemia, hypocalciuria, and elevated parathyroid hormone levels [1, 2]. FHH1 is caused by heterozygous mutations of the Calcium sensing-receptor (CaSR) [3, 4] and has a relatively benign course [5]. In stark contrast, homozygous mutations of CaSR cause neonatal severe hyperparathyroidism (NSHPT) which is associated with severe metabolic disturbances, growth retardation, and death [5, 6]. CaSR is expressed abundantly on the surface of parathyroid, thyroid and kidney cells [7, 8]. Activation of CaSR by increased extracellular $[Ca^{2+}]$ ($[Ca^{2+}]_o$) inhibits parathyroid hormone secretion, reduces bone resorption and renal tubular calcium reabsorption, maintaining $[Ca^{2+}]_o$ homeostasis [4, 9]. CaSR is a member of family 3 of the large G protein–coupled receptors (GPCRs) superfamily predicted to be 1,078 amino acids in length [10]. The first 19 amino acids form a signal peptide followed by a 593 amino acid extracellular domain (ECD), a 250 amino acid transmembrane domain (TMD), and a 216 amino acid intracellular tail [10]. Biochemical characterization of wild-type (WT) and mutant CaSR indicates dimerization is important for CaSR function [11, 12].

The majority of mutations that cause FHH1 and NSHPT are localized within the ECD, with the TMD representing the second most common site of mutation. Co-expression of reduced-function CaSR mutants with WT receptor reduced the sensitivity of populations of cells to external $[Ca^{2+}]$ [13–15]. The detailed mechanism by which this dominant negative causes FHH1 remains incompletely understood, but enhanced membrane trafficking of relevant mutants by following dimerization with WT CaSR is one hypothesis [16].

A well-studied, CaSR$^{-/-}$ mutant mouse deficient in exon 5 provides an excellent model of the human diseases FHH1 and NSHPT [17], although the pathophysiological mechanism remains unclear [18]. Characterization of the trafficking and signaling properties of the exon-5 deficient CaSR mutation (mCaSR) may help determine how dimerization of mutant and WT receptors contribute to CaSR signaling in FHH1 and NSHPT. Since CaSR heterodimerizes with other centrally-expressed GPCRs [19, 20], if mCaSR can still function after dimerizing with WT protein this may require reinterpretation of previous studies that employed the CaSR$^{-/-}$ mouse [21, 22]. Furthermore, the function of the exon-5 deficient CaSR mutation may have implications for understanding calcium regulation of epidermis differentiation [17]. Here we describe how heterologous expression of the WT and exon 5-deficient forms of the CaSR can be used to determine how mutant homodimers and heterodimers contribute to CaSR signaling in FHH1 and NSHPT.

## Materials and methods

### Plasmid construction

This study was approved by the Oregon Health & Science University Institutional Animal Care and Use Committee. Consent was not required as the study did not involve humans. Whole length EGFP-tagged CaSR was a kind gift from Dr. Emmanuel K. Awumey (North Carolina Central University) [23]. The vector containing mCaSR, pEGFP/mCaSR was obtained using the sense primer (5′- GTT GAG GCC TGG CAG GTG CCT TTC TCC -3′) and antisense primer (5′- GGA GAA AGG CAC CTG CCA GGC CTC AAC -3′)

by site-directed mutation according to manufacturer's protocol (QuikChange II XL Site-Directed Mutagenenesis Kit, 200521; Stratagene, LA Jolla, CA). The primers were designed to delete (in frame) 231 nucleotides (1378–1608) encoding exon 5 of CaSR. The mutation product was translated into a mutant receptor missing 77 amino acids (460–536) of CaSR. To obtain myc-tagged receptors, pcDNA-myc/CaSR and pcDNA-myc/mCaSR, BamHI sites were added at bases 603 of the vectors of pEGFP/CaSR and pEGFP/mCaSR by site-directed mutation using sense primer (CTA GCG CTA CCG GAT CCA GAT CTC GAG CTC) and antisense primer (GAG CTC GAG ATC TGG ATC CGG TAG CGC TAG) according to the manufacturer's protocol (QuikChange II). cDNAs encoding CaSR or mCaSR were obtained by BamHI digestion of pEGFP/CaSR and pEGFP/mCaSR and inserted into the BamH I site of pcDNA3.1/myc-His(-) A.

## Cell culture and transient transfection

Human embryonic kidney (HEK) 293 cells were maintained in Dulbecco's modified Eagle's medium supplemented with 5% fetal bovine serum (FBS). For $[Ca^{2+}]_i$ imaging and CaSR expression localization experiments, $10^5$ cells were suspended in 0.5 mL medium and placed onto a 12 mm diameter coverglass coated with 0.1 mg/ml poly-D-lysine. After one day in culture, cells were transfected with plasmid DNA following mixing with 50 μl Opti-MEM® I (reduced serum medium) and 2 μl Lipofectamine™ 2000 (Invitrogen). For Western blotting and co-immunoprecipitation experiments, $10^6$ cells were cultured in each well of 6-well plate. After one day in culture cells were transfected with a mix of plasmid DNA using 250 μl Opti-MEM® I reduced serum medium and 10 μl Lipofectamine™ 2000 (Invitrogen) per well. The transfection medium was replaced with DMEM containing 5% FBS 6 h later. The final DNA concentration in the transfection medium was 0.063 or 0.25 nM and cells were cultured for another day before use.

## Measurement of cell surface expression of CaSR or mCaSR

Cell surface expression of CaSR or mCaSR was measured by membrane EGFP fluorescence. CaSR-EGFP or mCaSR-EGFP expression was measured in HEK 293 cells transfected with plasmid. Briefly, cells were washed twice with Tyrode's solution composed of (in mM) 150 NaCl, 4 KCl, 1 CaCl$_2$, 1 MgCl$_2$, 10 HEPES, and 10 glucose, pH 7.35, and placed on the stage of an inverted microscope (OLYMPUS IX70) equipped with an 60x/1.42 oil-immersion objective (OLYMPUS PlanApo N). Cells were excited at 490 nm (Semrock BrightLine® FITC-3540B-OMF-ZERO Filter set) with a halogen light source (12V, 100W; OSRAM HLX XENO-PHOT 64625), and the emission at 510 nm was captured by a digital camera (HAMAMATSU ORCA-ER). The images were transmitted to computer and processed using Wasabi image software (Hamamatsu Photonics).

## Immunocytofluorescence

Cells growing on coverslips were transiently transfected with pcDNA-myc/CaSR, pcDNA-myc/mCaSR or pcDNA-myc/mCaSR and pEGFP/CaSR. After 24 hours, cells were washed twice with PBS, and fixed for 15 min with 4% (v/v) paraformaldehyde. Following fixation, cells were washed three times with PBS and blocked with PBS containing 1% bovine serum albumin and 2% normal goat serum for 30 min at room temperature, then incubated with 1:500 mouse anti-myc monoclonal antibody (Invitrogen) in the blocking solution overnight at 4˚C. The next day, cells were washed 3 times with PBS, then incubated with goat anti-mouse IgG Alexa-Fluor 594 (1:500), diluted in blocking solution for 60 min, and washed again 3 times with PBS.

Coverslips were then mounted in Vectashield reagent (Molecular Probes/Invitrogen, Eugene, OR). Images were acquired as above.

## Immunoprecipitation and western blot

Cells were lysed in ice-cold NET/T buffer composed of (in mM) 150 NaCl, 5 EDTA, 10 Tris, pH 7.4 with 1% Triton X-100 and 1x Complete Mini Protease Inhibitor Mixture (Roche Applied Science, Indianapolis, IN). Cell lysate was cleared by centrifugation at 15,000x g for 10 min. Protein concentration was measured using the BCA Protein Assay (Pierce). Lysates containing 200 μg of total protein were precleared with rec-protein A-Sepharose® 4B (Zymed Laboratories Inc., San Francisco, CA) beads for 60 min at 4°C. The precleared lysates were rotated overnight at 4°C with or without rabbit anti-EGFP antibody, and incubated with rec-protein A-Sepharose® 4B for 2 h. After centrifugation, the beads were washed three times with NET/T buffer, and the immunoprecipitated complexes were eluted with 2x Laemmli buffer [24] for 15 min at room temperature. Before loading on SDS-polyacrylamide gel electrophoresis (SDS-PAGE) gel, the proteins in cell lysate or the immunoprecipitated complexes were denatured by heating at 95°C for 5 min. All samples were subjected to 8% SDS-PAGE, followed by transfer to nitrocellulose, and immunodetection with mouse anti-EGFP (1:2,000) or anti-myc (1:5,000) primary monoclonal antibody, and horseradish peroxidase-conjugated secondary antibodies.

## Measurement of $[Ca^{2+}]_i$ response

$[Ca^{2+}]_i$ was measured in cells using the fluorescent indicator, X-rhod1. The relatively low affinity of X-rhod1 reduced saturation of the fluorescence signal in these experiments, and its red-shift permits clear separation of the $Ca^{2+}$ fluorescence from EGFP signals. Briefly, cells were loaded for 30 min at 37°C with 2 μM X-rhod1 in Tyrode's solution, and placed on the stage of an inverted microscope (OLYMPUS IX70) equipped with a 60x/1.42 oil-immersion objective (OLYMPUS PlanApo N). X-rhod-1 loaded cells were excited at 580 nm (Semrock BrightLine® TXRED-4040B-OMF-ZERO filter set) with halogen light source (an OSRAM HLX XENO-PHOT 64625 (12V, 100W)), and emission beyond 600 nm was captured by a digital camera every 2 second (HAMAMATSU ORCA-ER). The images were transmitted to computer and processed using Wasabi image software. The $[Ca^{2+}]_i$ values were reported as emission fluorescence ratios ($F/F_0$) of cells where $F_0$ was the baseline fluorescence before $[Ca^{2+}]_o$ change. About 20 cells with similar cell surface EGFP fluorescence from a single field were analyzed on each coverslip. Solutions were delivered though gravity-flow perfusion via a small dead space capillary and manifold. Data were analyzed using Igor Pro macros and concentration-effect relationships fitted with Eq 1:

$$(F - F_0)/F_0 = F_{max}/\left\{1 + \left(EC_{50}/\left[Ca^{2+}\right]_o\right)^n\right\} \tag{1}$$

where $F_{max}$, $EC_{50}$, and n represent the maximal $(F—F_0)/F_0$, the $[Ca^{2+}]_o$ activating half-maximal $(F—F_0)/F_0$, and the Hill coefficient respectively.

## Statistics

Significant differences between the values obtained in each assay from samples of cells transfected with various CaSR constructs were determined using the Student's t test and expressed as mean ± SEM. P values of less than 0.05 were considered significant.

## Results

### CaSR but not mCaSR is targeted to the plasma membrane

To examine the mechanisms by which CaSR mutations lead to FHH1 and NSHPT, we engineered CaSR and the exon-5 deficient CaSR mutant (mCaSR). Both constructs were then inserted into pEGFP and pcDNA-myc plasmids to code for fusion proteins (Fig 1A), and these were transiently transfected into HEK cells. CaSR exists as a homodimer on the cell surface and dimerization is not ligand-induced [11, 12]. Using confocal microscopy in live cells, we determined the EGFP-tagged wild-type CaSR was primarily expressed on the plasma membrane whereas mCaSR was distributed throughout the cytoplasm (Fig 1D–1I). In control experiments in untransfected HEK cells, plasma membrane was identified using FM1-43 and FM5-95 (Fig 1B, 1C), the green- and red-shifted amphiphilic styryl dyes that fluoresce when bound to phospholipid [25]. Peak signals from both dyes superimposed (Fig 1C) confirming minimal red-green refractive error within the imaging system. In transfected cells, fluorescence from CaSR-EGFP colocalized with FM5-95 fluorescence indicating that the majority of the CaSR was in the plasma membrane (Fig 1D–1G). In HEK cells transfected with EGFP-mCaSR alone, the EGFP signal overlapped minimally with the FM 5–95 indicating mutant protein was restricted mainly to the cytoplasm (Fig 1H, 1I).

### WT CaSR facilitated mutant CaSR traffic to cell membrane

CaSR forms homodimers in the endoplasmic reticulum and remains dimerized at the plasma membrane independent of ligand action [11, 12]. We hypothesized that mutant-WT CaSR heterodimer formation may rescue mCaSR trafficking and thereby enhance CaSR signaling in heterozygotes. To test this idea, we used myc- and EGFP-tagged forms of both WT and mCaSR (Fig 1A). CaSR-EGFP in live HEK cells and CaSR-myc in fixed HEK cells were targeted preferentially to the edge of the HEK cells (Fig 2A, top panels and Fig 1D). mCaSR-EGFP and mCaSR-myc were seen throughout the cytoplasm (Fig 2A, middle panel). In contrast, equimolar co-expression of mCaSR-EGFP with CaSR-myc resulted in mCaSR-EGFP clearly reaching the periphery of the cell (Fig 2A, lower panel). In addition, mCaSR-myc was similarly redirected towards the plasma membrane by CaSR-EGFP (Fig 2A, lower panel). These data indicate that there is rescue of the trafficking of mCaSR by co-expression of wild-type CaSR which may result from heterodimer formation (Fig 2B).

### The interaction between co-expressed CaSR and mCaSR

To test if WT CaSR and mCaSR interact, HEK cells were transfected with the mCaSR and/or CaSR constructs (Fig 1A), and then cell lysates were immunoprecipitated with anti-EGFP-coated beads. Beads were washed and then probed in Western blot analysis with antibodies against EGFP and myc tags (Fig 3). Wild-type CaSR-myc and CaSR-EGFP existed in immature forms (140 kDa) and mature forms (160 kDa) while mCaSR-myc and mCaSR-EGFP existed only in immature species of 150 kDa (Fig 3, right-hand lanes). CaSR-myc was pulled down by CaSR-EGFP (Fig 3, right upper section) confirming an interaction between these two forms of CaSR. Likewise, mCaSR-myc and CaSR-myc were immunoprecipitated by mCaSR-EGFP (Fig 3, right mid and lower, respectively) indicating mCaSR-mCaSR and mCaSR-CaSR interactions occurred. Similarly, interactions were detected when the cell lysates were tested without immunoprecipitation (Fig 3, middle lanes). These effects were not due to non-specific binding to the beads (Fig 3, left-hand lanes). These data indicate that wild-type and mutant CaSR bind, form multimers and thereby provide a mechanism by which WT CaSR may rescue trafficking of mCaSR.

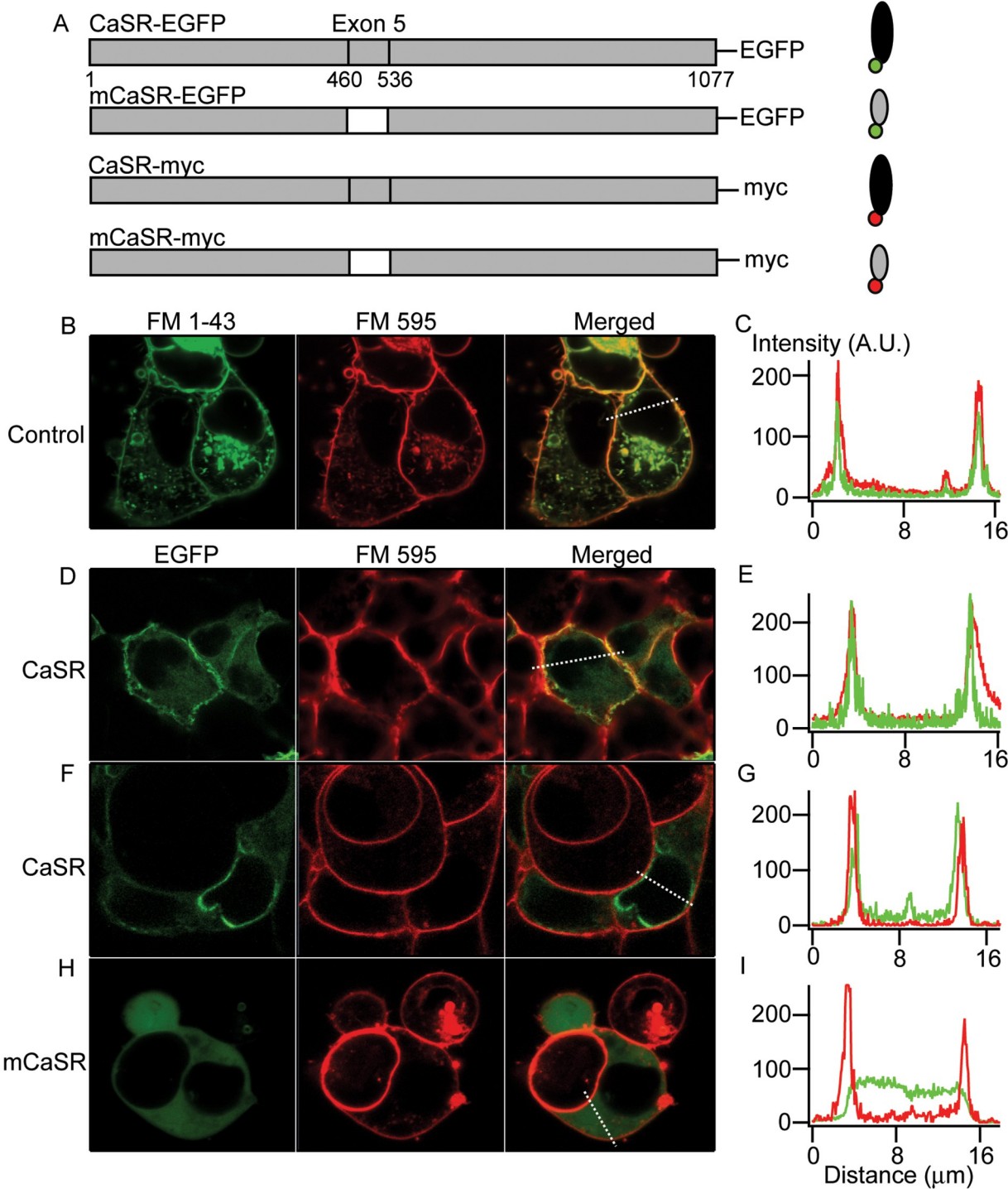

**Fig 1. CaSR but not mCaSR is trafficked to the cell membrane in HEK cells.** A, schematic representation of the CaSR constructs utilized. Deleted exon 5 was shown in white with construct sequence in gray. Numbers under diagrams for the full-length protein designate the first and last amino acid of CaSR (1–1077), and the first and last amino acid of CaSR exon 5 (460–536). EGFP or myc epitope tag was added to the C terminus of individual proteins. B, confocal images of living HEK cells co-labeled with red and green FM dyes, that fluoresce when bound to lipid, show superposition in the merged channel due to minimal refractive error. C, relative intensity of both fluorophores along line of interest (white broken line) in merged channel from B. D-G, two examples showing that the majority of CaSR-EGFP in HEK cells is sharply co-localized with FM-595 indicating it is targeted in or close to the cell membrane in confocal images. H, mCaSR is fairly uniformly distributed throughout the HEK cell with minimal overlap with FM-595 fluorescence. I, the line of interest indicates that the relative intensity of the EGFP signal is substantially reduced when the fluorescence from FM-595 increases indicating minimal mCaSR-EGFP in the cell membrane.

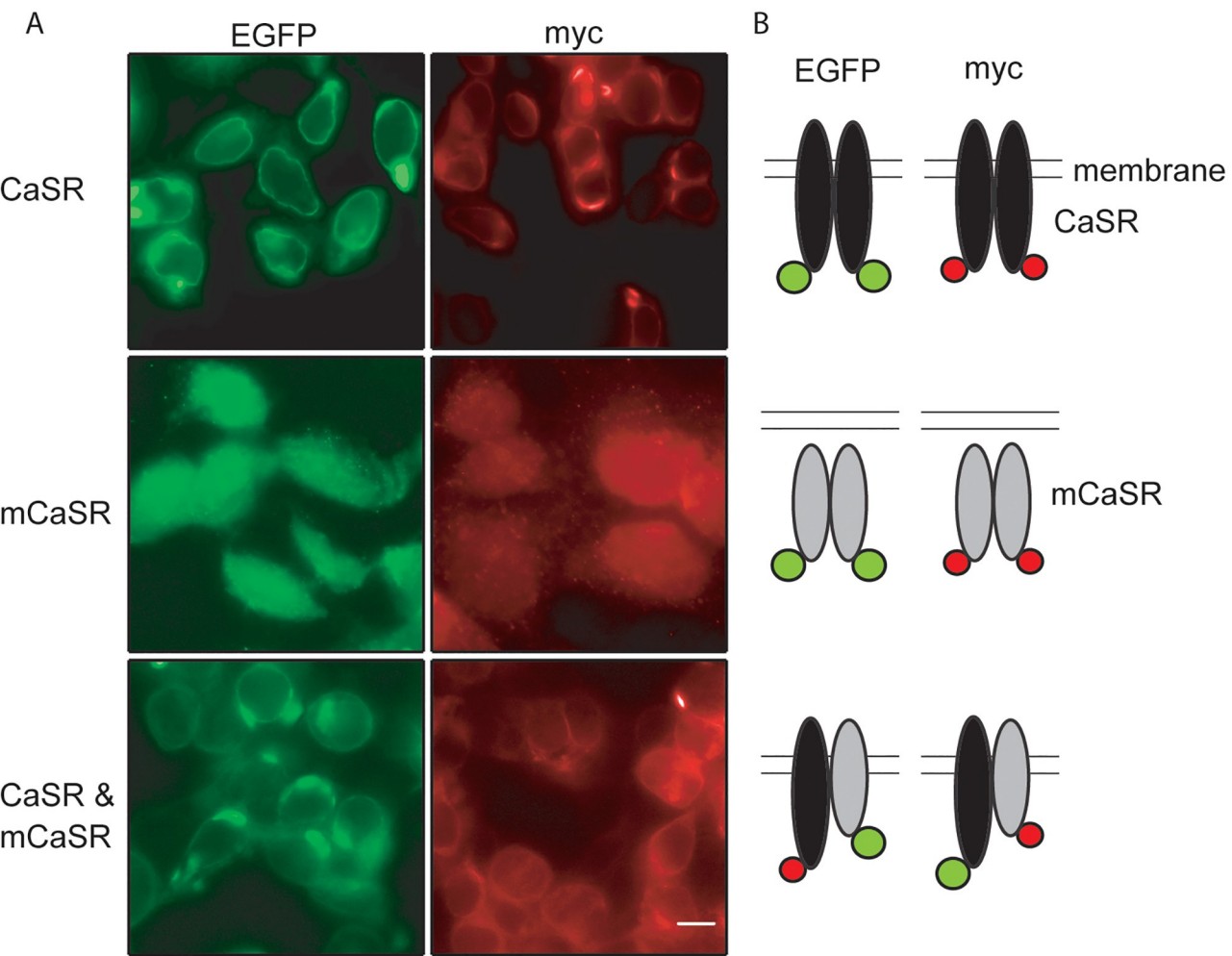

**Fig 2. CaSR-mCaSR heterodimers reach the cell membrane.** A, fluorescence images of HEK 293 cells transfected with CaSR and/or mCaSR fused with EGFP (left column) or myc epitope (right column). Cells were fixed and the myc tag detected using mouse anti-myc (1:500, Invitrogen) antibody and secondary goat anti-mouse antibody IgG AlexaFluor 594 (1:500). Scale bars = 10 μm. The fluorescence was confined to the cell perimeters when CaSR was expressed regardless of the tag identity (upper row) but the fluorescence was central if mCaSR was expressed alone (middle row). Peripheral concentration of the fluorescently tagged mCaSR was observed following co-expression of CaSR and mCaSR (lower row). B, Model illustrating wild-type (black) or mutant (gray) CaSR distributions relative to the plasma membrane are similar regardless of whether tagged with EGFP or myc. Co-expression with CaSR results in some mCaSR accumulating at the cell membrane.

## Mutant CaSR homomultimers do not detect changes in $[Ca^{2+}]_o$

While we found that mCaSR-EGFP and mCaSR-myc do not appreciably reach the cell membrane (Fig 1H, 1I), it is possible that mCaSR may be present at very low concentrations in the cell membrane and still able to detect changes in $[Ca^{2+}]_o$. We tested this hypothesis by measuring intracellular $[Ca^{2+}]$ in HEK cells that had been transiently transfected with both myc- and EGFP-tagged CaSR expression vectors, using the calcium sensitive fluorophore, X-rhod1. In HEK cells expressing WT CaSR, signaling results in the sequential activation of Gq [8], and phospholipase C (PLC) which increases production of inositol triphosphate (IP$_3$) and intracellular release of $Ca^{2+}$. Increasing the $[Ca^{2+}]_o$ of the perfusing solution from 1 to 10 mM (upper trace) reversibly increased $[Ca^{2+}]_i$ as evidenced by the increase in X-rhod1 fluorescence above basal (F/F$_0$) for both cells expressing both myc- and EGFP-tagged wild type CaSR (Fig 4A). In

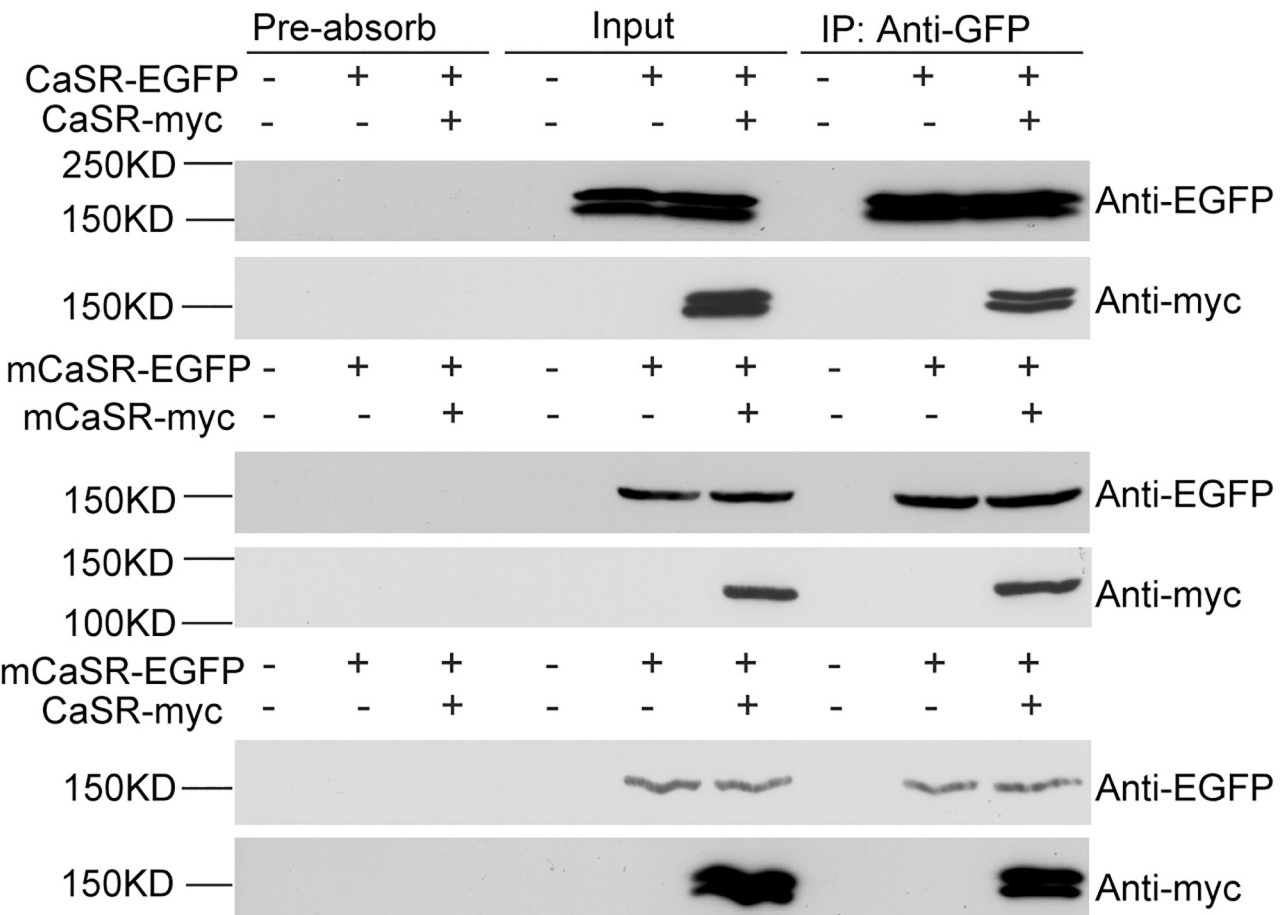

**Fig 3. Co-immunoprecipitation of CaSR and mCaSR following co-expression indicate they interact when co-expressed.** CaSR-EGFP or mCaSR-EGFP was immunoprecipitated with rabbit anti-GFP antibody. Co-immunoprecipitated proteins were detected with mouse anti-GFP and anti-myc monoclonal antibody. Pre-absorb lanes correspond to the lysates treated identically to immunoprecipitated samples but without anti-GFP antibody. These control lanes indicate that the observed interaction between CaSR, mCaSR, or CaSR and mCaSR is not due to nonspecific binding of either protein to rec-protein A-Sepharose® 4B beads.

the experiments illustrated in Fig 4A, $F/F_0$ peaked at $1.86 \pm 0.06$ and $1.85 \pm 0.04$ in CaSR-EGFP (n = 11) and CaSR-myc (n = 15) expressing cells respectively, <55 seconds after solution change and then decreased slowly during the application (Fig 4A, individual and average responses of 11 and 27 cells represented by gray and black traces, respectively). Cells express-ing mCaSR alone showed a delayed and markedly reduced response to extracellular calcium, indistinguishable from the response of untransfected HEK cells (Fig 4A; $F/F_0$ increased by $0.036 \pm 0.003$ (n = 27) for mCaSR, vs. $0.037 \pm 0.003$ (n = 22) for untransfected HEK cells). The change in $F/F_0$ during the $[Ca^{2+}]_o$ step was used to determine the concentration-effect rela-tionships for the previously described combinations of wild-type and mutant forms of the CaSR (Fig 4B). Each coverslip of cells was tested with a single $[Ca^{2+}]_o$ step to minimize varia-tion due to desensitization [23]. The $EC_{50}$ and maximum response were $4.6 \pm 0.2$ mM and $1.00 \pm 0.02$ arbitrary units (a.u.) and $4.6 \pm 0.2$ mM and $0.98 \pm 0.02$ a.u. for EGFP- and myc-tagged CaSR respectively. The responses of cells expressing mCaSR alone were indistinguish-able from untransfected cells (Fig 4B), which is consistent with the trafficking experiments that indicated mCaSR did not reach the cell membrane alone (Fig 2). Co-expression of mutant

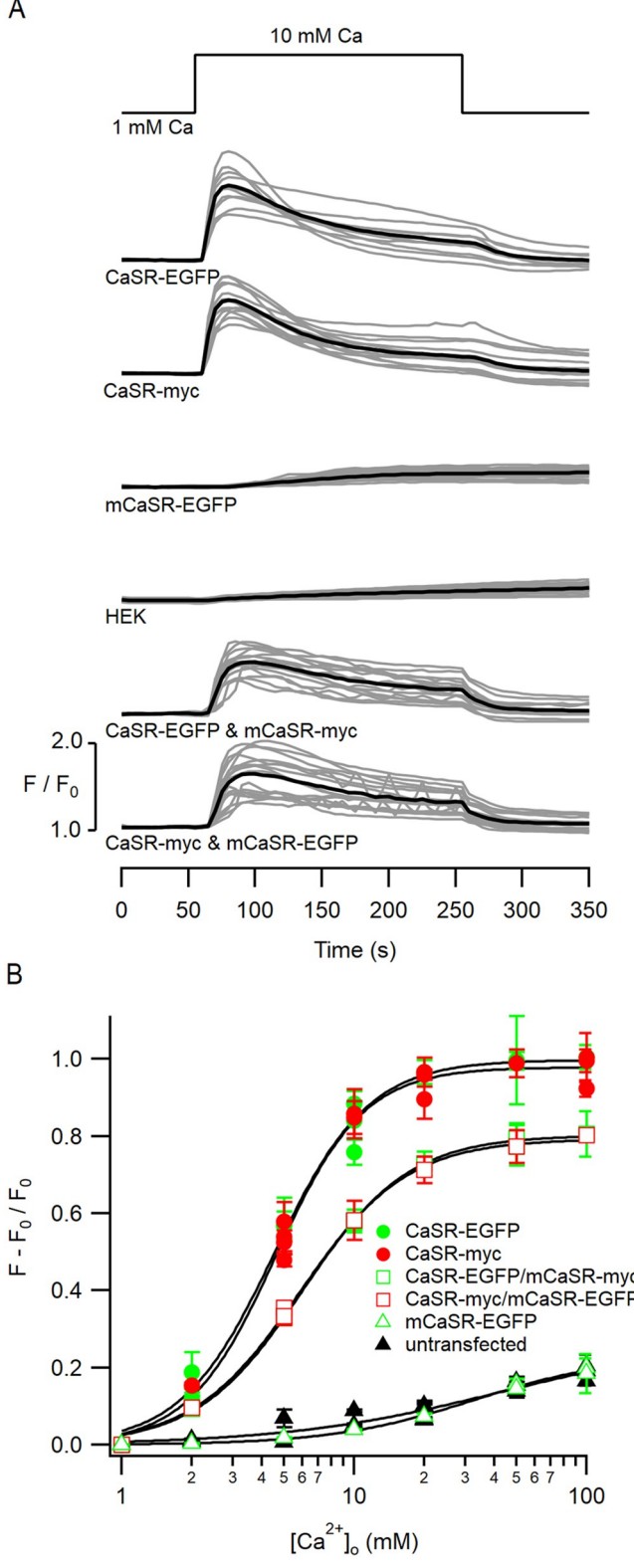

**Fig 4. $[Ca^{2+}]_o$-effect relationship for wild-type and mutant CaSR expressed alone or together.** A, the time courses of increases in $[Ca^{2+}]_i$, following the increase of $[Ca^{2+}]_o$ from 1 to 10 mM, in HEK cells transfected with pEGFP/CaSR, pcDNA-myc/CaSR, pEGFP/mCaSR, pEGFP/CaSR and pcDNA-myc/mCaSR, or pEGFP/mCaSR and pcDNA-myc/CaSR or untransfected controls. $[Ca^{2+}]_i$ changes, shown as $F/F_0$, was measured in cells grown on glass coverslips and

loaded with 2 μM X-rhod1 in Tyrode's solution. The gray traces are fluorescent signals for single cells in the field of view, while the average values (black) indicate the mean responses (n = 12–27). B, concentration-response relationship of $[Ca^{2+}]_o$-dependent mobilization of $[Ca^{2+}]_i$ in HEK 293 cells expressing wild-type and/or mutant CaSR. Each point reflects the average peak response above $F_0$ (see A) from a single coverslip and three separate coverslips were used for each $[Ca^{2+}]_o$ value. Values are means ± SEM. Curves were drawn according to equation1: for CaSR-EGFP, CaSR-myc, CaSR-EGFP/mCaSR-myc, mCaSR-EGFP/CaSR-myc, untransfected HEK, mCaSR-EGFP respectively: $F_{max}$ = 0.998 ± 0.021, n = 2.163 ± 0.171, and $EC_{50}$ = 4.552 ± 0.218; $F_{max}$ = 0.979 ± 0.015, n = 2.31 ± 0.231, and $EC_{50}$ = 4.603 ± 0.154; $F_{max}$ = 0.803 ± 0.008, n = 1.910 ± 0.086, and $EC_{50}$ = 6.021 ± 0.153; $F_{max}$ = 0.793 ± 0.012, n = 1.889 ± 0.114, and $EC_{50}$ = 5.845 ± 0.171; $F_{max}$ = 0.274 ± 0.089, n = 0.931 ± 0.266, and $EC_{50}$ = 42.722 ± 33.6; $F_{max}$ = 0.162 ± 0.010, n = 2.039 ± 0.43, and $EC_{50}$ = 19.195 ± 2.49.

and wild-type CaSR at the same total dose of plasmid DNA resulted in $EC_{50}$ and maximum response of 5.8 ± 0.2 mM and 0.79 ± 0.01 a.u. for mCaSR-EGFP-CaSR-myc and 6.0 ± 0.2 mM and 0.80 ± 0.01 a.u. for mCaSR-myc-CaSR-EGFP respectively. These data indicate that the myc and EGFP tags do not significantly affect CaSR signaling, that mCaSR homodimers do not transduce changes in $[Ca^{2+}]_o$, and that HEK cells co-expressing mutant and WT CaSR are less sensitive to changes in $[Ca^{2+}]_o$ than those transfected with WT alone.

## CaSR membrane levels determined by random association

The reduced sensitivity of the heterodimer expressing cells to external calcium is consistent with previous findings [15]. However, it is unclear whether this reflects reduced trafficking of the receptor to the membrane, or whether the reduced sensitivity is due to reduced function of the heterodimer compared to WT homodimer. Consequently, we next estimated the relative amounts of CaSR that reached the membrane under these conditions by using the EGFP tag and combinations of our CaSR mutants. CaSR-EGFP levels were measured in transfected cells to be used in $[Ca^{2+}]_i$ imaging experiments. Consistent with earlier findings, transfected cells were identified by the presence of the CaSR-EGFP fusion protein (Fig 5A). WT-CaSR-EGFP was restricted mainly at the edges of the HEK cells whereas the mCaSR-EGFP was present throughout the cytoplasm, and co-expression of both WT CaSR-myc and mCaSR-EGFP produced a mixed picture (Fig 5A, 5B). The peak values at the edges of the cells, representing membrane-associated EGFP-tagged protein, was measured under equivalent excitation conditions in 5–10 cells per coverslip and averaged over 3 coverslips that had undergone similar transfections (see Methods). CaSR production and trafficking appeared proportional to the amounts of transfected plasmid DNA; the fluorescence intensity in HEK 293 cells transfected with 250 pM pEGFP/CaSR was approximately four-fold higher than those transfected with 62.5 pM (25% CaSR-EGFP; Fig 5C). Transfection of 125 pM WT CaSR-EGFP with 125 pM mCaSR-myc produced a peak membrane fluorescence 51% of the 250 pM CaSR-EGFP transfection (Fig 5C) and equivalent to that predicted by random association of WT CaSR and mCaSR (Fig 5D). The average peak membrane fluorescence was reduced by a further 58% when equivalent amounts of WT CaSR-myc and mCaSR-EGFP were co-transfected, which was close to that predicted by random association (Fig 5C, 5D). Expression of mutant CaSR-EGFP alone was localized mainly in the cytoplasm and these peak values were only 13% of the values measured at the edges of cells transfected with WT CaSR-EGFP (Fig 5). There was minimal contaminating signal from cells on coverslips transfected with myc-tagged WT CaSR nor mCaSR (Fig 5C). These data indicate that in HEK cells, WT CaSR heterodimerization with mCaSR can direct mCaSR to the cell membrane. Taken together these findings suggest that the trafficking defect of mCaSR is completely rescued by WT CaSR association.

In the absence of compensatory mechanisms, co-expression WT and mCaSR will result in failure of mCaSR homodimer pairs to reach the plasma membrane, reducing by 25% both the

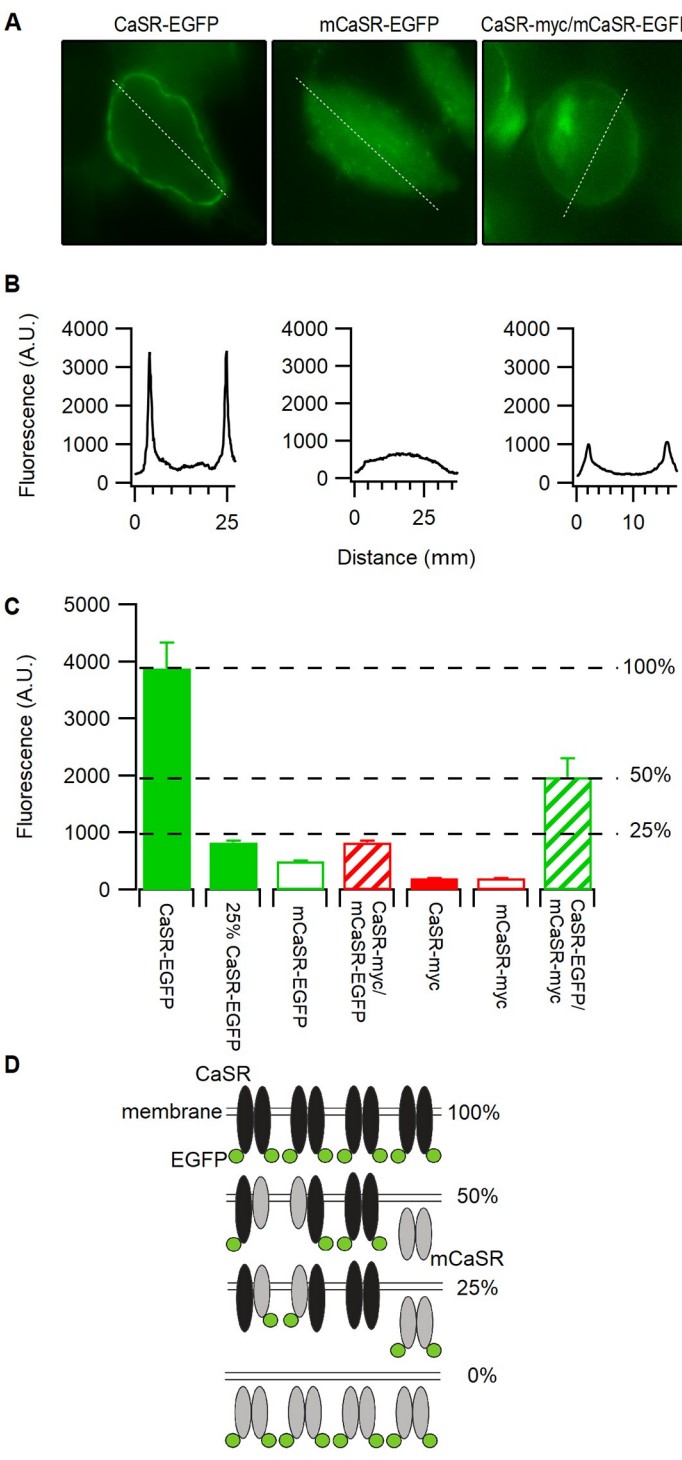

**Fig 5. Cell surface expression levels of wild-type and mutant CaSR.** A, EGFP fluorescence of HEK 293 cells transfected with pEGFP/CaSR, pEGFP/mCaSR, or pEGFP/mCaSR and pcDNA-myc/CaSR. The white line across the cell center and membrane is used to measure the receptor expression tagged with EGFP fluorescence. B, EGFP fluorescence analyzed using Wasabi image software to measure the fluorescence along the line of interest (broken white line) shown in A. The peak fluorescence was used as the cell surface expression of EGFP tagged receptor. C, cell surface receptors in HEK 293 cells transfected with 0.25 nM or 0.06 nM vector using the combinations shown in the Figure. By tagging either the wild-type or mutant receptors we measured relative efficiencies of expression of the wild-type and mutant CaSR in the membrane. Each histogram bar represents the average measurements from three separate coverslips (approximately 20 cells on each coverslip). Values are means ± SEM. D, Model illustrating that EGFP

membrane fluorescence of 25% of control when the mutant CaSR is EGFP tagged and 50% of control when the wild-type CaSR is tagged is consistent with mCaSR membrane expression being fully rescued by dimerization with wild-type CaSR in co-expression transfections.

total amount of CaSR localized at the membrane and the maximal effect of CaSR signaling (Figs 4B, 5D). The apparent increase in $EC_{50}$ of HEK cells co-expressing WT- and mCaSR relative to cells transfected with WT CaSR alone (6 mM vs 4.6 mM) led us to hypothesize that CaSR heterodimers have a lower affinity for $Ca^{2+}$ than WT homodimers. We tested this idea by examining the function of the various CaSR receptor isoforms under conditions designed to minimize desensitization and allow comparison. Groups (10–20) of transfected cells were identified by the presence of EGFP then perfused with a solution containing 0.2 mM $Ca^{2+}$ before switching to a solution containing 0.5–50 mM $Ca^{2+}$ for 100 sec and then back to 0.2 mM $Ca^{2+}$ for 10 mins. The low $[Ca^{2+}]_o$ and ten-minute wash were utilized to reduce basal desensitization and this allowed us to challenge each group of cells with three test $[Ca^{2+}]_o$. The peak $F/F_0$ was measured in individual cells and the average for each coverslip (n = 10–20) used to derive the concentration-effect relationship for the untransfected cells and cells transfected with WT, or WT and mutant CaSR (Fig 6A). Each $Ca^{2+}$ concentration is represented by the response of 2–4 coverslips and these data were fitted to the Hill equation (Fig 6A). To isolate the functional properties of the CaSR isotypes we subtracted the average response of untransfected cells from the average responses of the transfected cells. The subtracted responses of cells transfected with WT CaSR-EGFP (62.5 pM and 250 pM) are replotted in Fig 6B and both fitted with the Hill equation. The $EC_{50}$ of two fits were very similar at 2.43 ± 0.16 and 2.27 ± 0.09 mM whereas the maximal values were 0.227 ± 0.005 and 0.879 ± 0.013 for the 62.5 pM (25%) and 250 pM (100%) WT CaSR-EGFP groups respectively. The linear dependence of WT CaSR function on CaSR transfection levels was also illustrated by scaling the curve derived from the 100% group by 0.25. The resulting curve (red broken) was almost indistinguishable from the black curve fit to the data confirming that transfection is proportional to expression under these conditions.

Next, we used a similar subtraction method to determine the response of heterodimers expressed in HEK cells transfected with 125 pM WT CaSR and mCaSR. Our previous data predicted that the response of these cells was comprised of the signals from heterodimers, half as many WT homodimers, and the response of mutant homodimers that was equivalent to untransfected HEK cells. The heterodimer-dependent fraction (red closed squares) should therefore equal the response of the cells transfected with 125 pM of WT CaSR and 125 pM mCaSR (red open squares) minus the response of cells transfected with 62.5 pM WT CaSR (open green circles; Fig 6A, 6C). The Hill equation fit to the heterodimer signal revealed a three-fold lower affinity for $[Ca^{2+}]_o$ ($EC_{50}$ 7.15 ± 0.64 mM) compared with WT and a maximal value of 0.396 ± 0.19 close to the value predicted when the 100% homodimer curve was scaled by 0.5 to account for lower expression levels (scaled heterodimer; broken red curve). In summary, the maximal $F/F_0$ response was dependent on CaSR membrane level- WT CaSR homo- or WT CaSR-mCaSR heterodimers producing the same maximal effect and mCaSR-mCaSR dimers mediating no signal. However, the affinity of WT CaSR was three times greater than that of the CaSR-mCaSR heterodimer (Fig 6C). These differences in trafficking and affinity may underlie the severity of the clinical differences between FHH1 and NSHPT.

## Discussion

The CaSR is a major player in the physiology of $[Ca^{2+}]_o$ homeostasis and reduced function mutations cause mild or severe disease states depending on whether expression is hetero- or

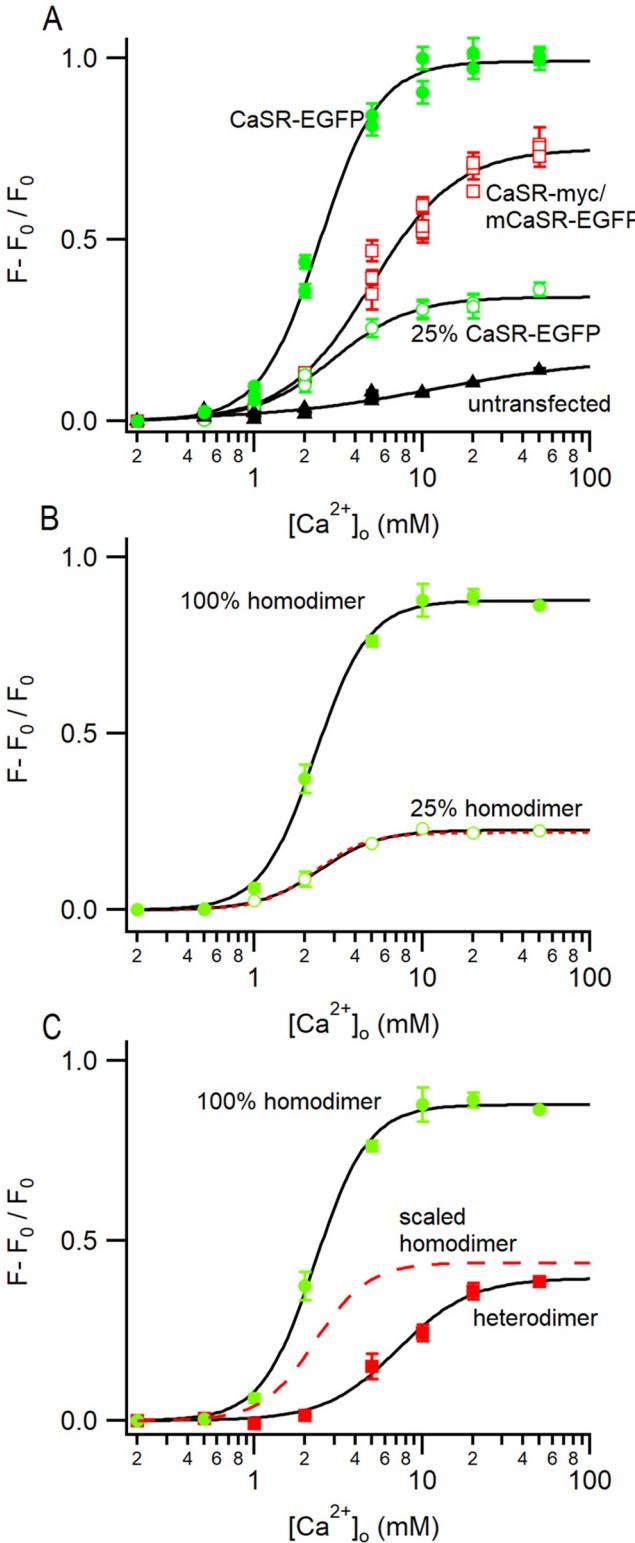

**Fig 6. WT/mutant CaSR heterodimers have a lower affinity but similar efficacy as WT CaSR.** A, concentration-response relationship of $[Ca^{2+}]_o$ dependent mobilization of $[Ca^{2+}]_i$ in HEK 293 cells non-transfected or transfected with 0.25 nM pEGFP/CaSR, 0.125 nM pEGFP/mCaSR and 0.125 nM pcDNA-myc/CaSR, or 0.06 nM pEGFP/CaSR. $[Ca^{2+}]_i$ changes, $((F-F_0) / F_0)$ was measured in cells loaded with 2 μM X-rhod1 following the increase in $[Ca^{2+}]_o$ from 0.2 mM. Differences between basal and peak $[Ca^{2+}]_i$ were plotted against $[Ca^{2+}]_o$. Values are means ± SEM and fitted

to Eq 1 for CaSR-EGFP, CaSR-myc/mCaSR-EGFP, and 25% CaSR-EGFP respectively: $F_{max}$ = 0.99105 ± 0.0158, $EC_{50}$ = 2.452 ± 0.099, and n = 2.459 ± 0.184; $F_{max}$ = 0.750 ± 0.026, $EC_{50}$ = 4.966 ± 0.301, and n = 1.703 ± 0.139; $F_{max}$ = 0.341 ± 0.010, $EC_{50}$ = 2.955 ± 0.235, and n = 1.866 ± 0.223. B, $[Ca^{2+}]_o$ dependent mobilization of $[Ca^{2+}]_i$ mediated by WT CaSR shows the average from three coverslip data at each $[Ca^{2+}]_o$ (0.25 nM and 0.06 nM wild-type vector transfection reflect 100% and 25% wild-type CaSR function respectively) after subtraction of average data for untransfected cells (all from A). Black curves are fitted using Eq 1 to 100% and 25% homodimer respectively: $F_{max}$ = 0.877 ± 0.012, $EC_{50}$ = 2.294 ± 0.081, and n = 2.735 ± 0.231; $F_{max}$ = 0.226 ± 0.005, $EC_{50}$ = 2.442 ± 0.131, and n = 2.432 ± 0.255; Broken red curve is curve fitted to 100% homodimer and then scaled by 0.25. C, concentration-effect relationship of CaSR-mCaSR heterodimers estimated by subtracting 25% CaSR-EGFP (open green squares) from average CaSR-myc/CaSR-EGFP (open red squares, both in A). The red squares describe signaling attributable to the heterodimers compared to the 100% WT homodimer signal plotted in B. After scaling the homodimer curve (see B) by 0.5 to account for the lower expression levels (red broken curve), the maximum effect of the heterodimer is similar but the affinity for $[Ca^{2+}]_o$ is right-shifted indicating a lower affinity. Black curves are fitted using Eq 1 to heterodimer: $F_{max}$ = 0.396 ± 0.019, $EC_{50}$ = 7.149 ± 0.641, and n = 1.990 ± 0.295.

homozygous, respectively [4, 26]. We determined that the exon 5-deficient CaSR mutation, that underlies an accurate mouse model of FHH1 and NSHPT, has a number of properties that account for the large differences between the two diseases. We found that in HEK cells, mCaSR is not trafficked to the plasma membrane when expressed alone (Figs 1, 2) but that once co-expressed with wild-type CaSR it reaches the membrane in quantities explained by random association (Fig 5). In other words, the mutant-wild type heterodimers and WT homodimers are trafficked equally well. In parallel experiments examining the impact of the mutation on CaSR function, we determined that the reduced performance of the heterodimers is due to their lower affinity, compared to WT homodimers, whereas the efficacies of both dimer forms are equal (Fig 6).

Extrapolation of these findings to the *in vivo* situation indicates that NSHPT would result from a complete loss of expression of CaSR at the plasma membrane whereas in FHH1 there would be in total 75% of normal levels of CaSR (WT and mutant), 67% of which would have a three-fold reduced $[Ca^{2+}]_o$ affinity compared to normal. In patients, the impact of these changes may well be mitigated by upregulation of expression of the CaSR, heterodimerization of the CaSR with other GPCRs, or other compensatory mechanisms. Evidence for potential compensatory mechanisms, arise from studies that have identified heterozygous mutations of G-protein subunit $\alpha_{11}$ (G$\alpha_{11}$) and adaptor protein 2 (AP2) as causes for diseases FHH2 and FHH3 respectively [4, 27]. G$\alpha_{11}$ mutations impair coupling with the upstream CaSR or the downstream phospholipase C, whereas AP2 mutations impair CaSR trafficking [4, 27, 28]. The importance of the multiple components in the process illustrates the complexity of calcium homeostasis.

CaSR heterodimerizes with other GPCRs under physiological conditions [20], raising the possibility that the trafficking and/or signaling of mutant CaSR could also be partially rescued by GPCRs in other contexts [19]. Such a mechanism might also confound experiments utilizing the exon 5-deficient mutant mouse where the mCaSR might reach the membrane and mediate the normal CaSR functions with partial potency and attenuate the impact of the "null-mutant". In the brain, CaSR signaling has been shown to cluster at nerve terminals [29] and regulate spontaneous and evoked transmission [21, 22, 30]. Interactions between CaSR and other GPCRs, such as the GABA$_B$ receptor, occur in intact brain [31, 32] indicating rescue by heterodimerization may occur and attenuate the apparent contribution of CaSR to signaling in the nervous system. However, such an effect does not account for the recent observation that CaSR does not regulate a non-selective cation channel and thereby control calcium-dependent excitability in the neocortex [33], since these studies employed a CaSR null-mutant with no residual membrane fragment [34].

The $EC_{50}$ for the WT CaSR seen here is similar to those previously reported [13–16]. The small differences in the Hill coefficient may reflect our use of lower affinity fluorophores in an attempt to minimize non-linearities from fluorophore saturation or possibly be due to our use of single cell fluorescence measurements to minimize differences arising from heterogeneous cell populations [35, 36].

## Conclusion

In summary, this study showed that exon 5-deleted mutant CaSR, which causes NSHPT in mice when homozygously expressed, showed no functional cell surface expression and lost its ability to sense $[Ca^{2+}]_o$ change in HEK 293 cells. Mutant CaSR trafficking to the cell surface was completely rescued by heterodimerization with WT CaSR. CaSR/mCaSR heterodimers showed similar efficacy and decreased affinity compared with WT CaSR homodimers. Overall, decreased cell surface expression and affinity of WT/mutant CaSR heterodimers combine to decrease the CaSR signaling that causes FHH1.

## Supporting information

**S1 Fig.**
(PDF)

## Acknowledgments

Thanks to Dr. Emmanuel K. Awumey from North Carolina Central University for his kind gift of WT rat CaSR construct and to Mr Luke Steiger and Dr Salil Rajayer for their comments on an earlier version of the manuscript.

## Author Contributions

**Conceptualization:** Stephen M. Smith.

**Data curation:** Stephen M. Smith.

**Formal analysis:** Xiaohua Wang, Stephen M. Smith.

**Funding acquisition:** Stephen M. Smith.

**Investigation:** Xiaohua Wang.

**Methodology:** Xiaohua Wang, James Lundblad, Stephen M. Smith.

**Project administration:** Xiaohua Wang, Stephen M. Smith.

**Supervision:** James Lundblad, Stephen M. Smith.

**Writing – original draft:** Xiaohua Wang.

**Writing – review & editing:** James Lundblad, Stephen M. Smith.

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
