## [Decision Letter · Decision Letter 0]

25 May 2022

PONE-D-22-09433Reduced affinity of calcium sensing-receptor heterodimers and reduced mutant homodimer trafficking combine to impair function in a mouse model of Familial Hypocalciuric Hypercalcemia Type 1PLOS ONE

Dear Dr. Smith,

Thank you for submitting your manuscript to PLOS ONE. After careful consideration, we feel that it has merit but does not fully meet PLOS ONE’s publication criteria as it currently stands. Therefore, we invite you to submit a revised version of the manuscript that addresses the points raised during the review process.

We look forward to receiving your revised manuscript.

Kind regards,

Xiangming Zha, Ph.D.

Academic Editor

PLOS ONE

Journal Requirements:

2. During your revisions, please confirm whether the wording in the title is correct and update it in the manuscript file and online submission information if needed. Specifically, the title refers to a mouse model but the manuscript refers to experiments in human HEK cell lines. Please keep in mind that conclusions (including inferences in the title) should be presented appropriately and supported by the results.

[Thanks to Dr. Emmanuel K. Awumey from North Carolina Central University for his kind gift of WT rat CaSR construct. Research reported in this publication was supported by the National Institute of General Medical Sciences of the National Institutes of Health (R01GM134110) and by U.S. Department of Veterans Affairs (BX002547). The content is solely the responsibility of the authors and does not necessarily represent the official views of the National Institutes of Health or the U.S. Department of Veterans Affairs.]

 [ Research reported in this publication was supported by the National Institute of General Medical Sciences of the National Institutes of Health (R01GM134110) and by U.S. Department of Veterans Affairs (BX002547) to SMS. The content is solely the responsibility of the authors and does not necessarily represent the official views of the National Institutes of Health or the U.S. Department of Veterans Affairs. The funders had no role in study design, data collection and analysis, decision to publish, or preparation of the manuscript.]

Reviewers' comments:

Reviewer's Responses to Questions

**Comments to the Author**

1. Is the manuscript technically sound, and do the data support the conclusions?

Reviewer #1: Partly

Reviewer #2: Yes

2. Has the statistical analysis been performed appropriately and rigorously? 

Reviewer #1: Yes

Reviewer #2: Yes

3. Have the authors made all data underlying the findings in their manuscript fully available?

Reviewer #1: Yes

Reviewer #2: Yes

4. Is the manuscript presented in an intelligible fashion and written in standard English?

Reviewer #1: Yes

Reviewer #2: Yes

5. Review Comments to the Author

Reviewer #1: The authors studied a model of familial hypocalciuric hypercalcemia type 1. The model consists of transfecting and co-transfecting the wild type (WT) and mutated (mCaSR) in human embryonic kidney (HEK) cells.

In the abstract, the authors claim that mCaSR homodimer is localized to the cytoplasm and lack function, the functional WT CaSR homodimer is localized to the membrane, and the heterodimer is localized to the membrane but has decreased sensitivity to extracellular Ca2+ ions.

The data supports the paper’s conclusions as far as the results in the HEK cell line. The data in Figure 5 with different transfection percentage of wild type and mutated CaSR would be stronger if the author demonstrated that the protein levels of CaSR and mCaSR correspond to the transfected DNA levels, as it is often not the case (for example, mutants are often targeted for degradation more than wild type proteins).

The more precise title to the paper would be “Reduced affinity of Ca2+-sensing wild type-mutant heterodimers and reduced mutant homodimer trafficking combine to impair Ca2+ sensing by Ca2+-sensing receptor with the exon 5-deleted mutation”, as the current title claims results in the mouse model though data were obtained in a cell line, and the existence of the same mechanism in the mouse model still needs to be shown. The wording in the abstract and the conclusions also have to be adjusted according to what was actually shown in the paper.

Minor points:

Line 33: a space between “expressing” and “mutant” is missing.

Line 78: “in”?

Line 87: “to” is missing.

Line 109: “Human”.

Line 113: full stop is missing.

Line 117: extra “in”.

Line 207: Is F-G panels are another example similar to D-E?

Reviewer #2: The authors successfully showed that exon 5-deleted mutant CaSR, which causes NSHPT in mice when homozygously expressed, showed no functional cell surface expression and lost its ability to sense [Ca2+]o change in HEK 293 cells. They also showed that mutant CaSR trafficking to the cell surface could be rescued by heterodimerization with WT CaSR.

The result looks promising, and I hope they will pursue their findings in animal studies to combat CaSR mutation-related diseases.

Minor correction:

1. Page 11, line 205-207: Clarify figure 1F,G.

2. Figure 1: D-E and F-G should explain clearly in the figure legend.

6. PLOS authors have the option to publish the peer review history of their article (what does this mean?). If published, this will include your full peer review and any attached files.

Reviewer #1: No

Reviewer #2: No

---

## [Author Response · Author response to Decision Letter 0]

16 Jun 2022

I write to resubmit our manuscript [PONE-D-22-09433] after responding to the reviewers and modifying in accordance with the journal requirements.

We thank the reviewers for their careful reading of our manuscript and believe that the document has been improved after their helpful reviews. Please find a point by point below. In addition, we have altered the title, acknowledgements, and figures in accordance with the suggestions. Please do not change the Funding Statement. The unadjusted blot images are submitted as a supplementary information file. Patients and animals were not utilized in this study which was approved by the OHSU IACUC and OHSU safety committee. 

Detailed response to Reviewers:

Reviewer #1: The authors studied a model of familial hypocalciuric hypercalcemia type 1. The model consists of transfecting and co-transfecting the wild type (WT) and mutated (mCaSR) in human embryonic kidney (HEK) cells.

In the abstract, the authors claim that mCaSR homodimer is localized to the cytoplasm and lack function, the functional WT CaSR homodimer is localized to the membrane, and the heterodimer is localized to the membrane but has decreased sensitivity to extracellular Ca2+ ions.

The data supports the paper’s conclusions as far as the results in the HEK cell line. The data in Figure 5 with different transfection percentage of wild type and mutated CaSR would be stronger if the author demonstrated that the protein levels of CaSR and mCaSR correspond to the transfected DNA levels, as it is often not the case (for example, mutants are often targeted for degradation more than wild type proteins).

The reliable direct measurement of protein levels in single cells is difficult if one is trying to assay only that reaching the plasma membrane. Measurements of protein in populations of cells may result in considerable variance due to inclusion of transfected and untransfected cells. We attempted to measure the protein levels at the membrane by tagging the WT or mCaSR with EGFP and employing the fluorescence as a measure of the amount of protein reaching the membrane (Fig 5 C). We were able to do this as using 25% of the WT DNA resulted in ~25% of the EGFP at the membrane. A similar level of fluorescence (~25%) was detected at the membrane when half of the DNA was mCaSR-EGFP and half was CaSR-myc, consistent with the expectation shown in the model (Fig 5D, 3rd row). The small amounts of fluorescence measured when DNA for mCaSR-EGFP alone was used was due to scattered light from protein confined to the cytoplasm (see Fig 1I). 

The more precise title to the paper would be “Reduced affinity of Ca2+-sensing wild type-mutant heterodimers and reduced mutant homodimer trafficking combine to impair Ca2+ sensing by Ca2+-sensing receptor with the exon 5-deleted mutation”, as the current title claims results in the mouse model though data were obtained in a cell line, and the existence of the same mechanism in the mouse model still needs to be shown. The wording in the abstract and the conclusions also have to be adjusted according to what was actually shown in the paper.

We agree that the title was misleading and have changed it to:

Reduced affinity of calcium sensing-receptor heterodimers and reduced mutant homodimer trafficking combine to impair function in a model of familial hypocalciuric hypercalcemia type 1.

Minor points:

Line 33: a space between “expressing” and “mutant” is missing. 

Corrected.

Line 78: “in”?

Corrected.

Line 87: “to” is missing.

Corrected.

Line 109: “Human”.

Corrected.

Line 113: full stop is missing.

Corrected.

Line 117: extra “in”.

Corrected.

Line 207: Is F-G panels are another example similar to D-E?

The F-G panels are examples similar to D-E but provide examples of the variability across a culture. I have modified the legend to make this clear. 

Reviewer #2: The authors successfully showed that exon 5-deleted mutant CaSR, which causes NSHPT in mice when homozygously expressed, showed no functional cell surface expression and lost its ability to sense [Ca2+]o change in HEK 293 cells. They also showed that mutant CaSR trafficking to the cell surface could be rescued by heterodimerization with WT CaSR.

The result looks promising, and I hope they will pursue their findings in animal studies to combat CaSR mutation-related diseases.

We are pleased that you found our results promising. 

Minor correction:

1. Page 11, line 205-207: Clarify figure 1F,G. 

The F-G panels are examples similar to D-E but provide examples of the variability across a culture. I have modified the text to include these panels and the legend to make this clear. 

2. Figure 1: D-E and F-G should explain clearly in the figure legend.

Please see response to question 1.

---

## [Editor Report · Decision Letter 1]

1 Jul 2022

Reduced affinity of calcium sensing-receptor heterodimers and reduced mutant homodimer trafficking combine to impair function in a model of familial hypocalciuric hypercalcemia type 1

PONE-D-22-09433R1

Dear Dr. Smith,

We’re pleased to inform you that your manuscript has been judged scientifically suitable for publication and will be formally accepted for publication once it meets all outstanding technical requirements.

Kind regards,

Xiangming Zha, Ph.D.

Academic Editor

PLOS ONE
---

## [Editor Report · Acceptance letter]

11 Jul 2022

PONE-D-22-09433R1 

Reduced affinity of calcium sensing-receptor heterodimers and reduced mutant homodimer trafficking combine to impair function in a model of familial hypocalciuric hypercalcemia type 1 

Dear Dr. Smith:

I'm pleased to inform you that your manuscript has been deemed suitable for publication in PLOS ONE. Congratulations! Your manuscript is now with our production department. 

Kind regards, 

on behalf of

Dr. Xiangming Zha 

Academic Editor

PLOS ONE